# Real World Cost-Effectiveness Analysis of Population Screening for BRCA Variants among Ashkenazi Jews Compared with Family History-Based Strategies

**DOI:** 10.3390/cancers14246113

**Published:** 2022-12-12

**Authors:** Rachel Michaelson-Cohen, Matan J. Cohen, Carmit Cohen, Dan Greenberg, Amir Shmueli, Sari Lieberman, Ariela Tomer, Ephrat Levy-Lahad, Amnon Lahad

**Affiliations:** 1Medical Genetics Institute, Shaare Zedek Medical Center, Jerusalem 9112102, Israel; 2Faculty of Medicine, Hebrew University of Jerusalem, Jerusalem 9112102, Israel; 3Clalit Health Services, Jerusalem District, Jerusalem 9112102, Israel; 4Department of Health Policy and Management, School of Public Health, Faculty of Health Sciences, Guilford Glazer Faculty of Business and Management, Ben-Gurion University of the Negev, Be’er-Sheva 8410501, Israel; 5Department of Health Management & Economics, School of Public Health, Hebrew University of Jerusalem, Jerusalem 9112102, Israel

**Keywords:** BRCA, breast cancer, ovarian cancer, cancer risk-reduction, NGS, molecular genetic testing, population screening, cost-effectiveness analysis, economic evaluation

## Abstract

**Simple Summary:**

Identifying BRCA mutations carriers reduces cancer incidence by surveillance and prevention. We analyzed the cost-effectiveness of population screening (PS) for BRCA mutations in Ashkenazi Jews (AJ), for whom carrier rate is 2.5%, compared with existing strategies: cascade testing (CT) in carrier’s relatives, and international family history (IFH)-based guidelines. We estimated quality-adjusted life-years (QALYs) gained, and cost-effectiveness for PS vs. existing strategies. Per 1000 women, PS vs. CT predicted 21.6 QALYs gained, and lifetime decrease of three breast cancer (BC) and four ovarian cancer (OC) cases, and PS vs. IFH predicted 6.3 QALYs gained, decrease of 1 BC and 1 OC. PS was less costly than CT (−3097 USD/QALY), and more costly than IFH (+42,261 USD/QALY), yet still cost-effective, and the most effective screening strategy for cancer prevention. The alternative strategies restrict the number of carriers identified, precluding cancer prevention in unidentified carriers. Population BRCA testing should be available to all AJ women.

**Abstract:**

Identifying carriers of pathogenic BRCA1/BRCA2 variants reduces cancer morbidity and mortality through surveillance and prevention. We analyzed the cost-effectiveness of BRCA1/BRCA2 population screening (PS) in Ashkenazi Jews (AJ), for whom carrier rate is 2.5%, compared with two existing strategies: cascade testing (CT) in carrier’s relatives (≥25% carrier probability) and international family history (IFH)-based guidelines (>10% probability). We used a decision analytic-model to estimate quality-adjusted life-years (QALY) gained, and incremental cost-effectiveness ratio for PS vs. alternative strategies. Analysis was conducted from payer-perspective, based on actual costs. Per 1000 women, the model predicted 21.6 QALYs gained, a lifetime decrease of three breast cancer (BC) and four ovarian cancer (OC) cases for PS vs. CT, and 6.3 QALYs gained, a lifetime decrease of 1 BC and 1 OC cases comparing PS vs. IFH. PS was less costly compared with CT (−3097 USD/QALY), and more costly than IFH (+42,261 USD/QALY), yet still cost-effective, from a public health policy perspective. Our results are robust to sensitivity analysis; PS was the most effective strategy in all analyses. PS is highly cost-effective, and the most effective screening strategy for breast and ovarian cancer prevention. BRCA testing should be available to all AJ women, irrespective of family history.

## 1. Introduction

Breast and ovarian cancer are major health concerns. Breast cancer (BC) is the most common cancer and the leading cause of cancer death in Israeli women, while ovarian cancer (OC) is the most lethal gynecological cancer [1,2]. BC and OC combined were the leading overall cause of death in Israeli women in 2019 [1]. In Ashkenazi Jews (AJ), ~10% of BC and ~40% of OC are caused by variants in BRCA [3,4,5]. Among AJ, 1:40 is a carrier of one of three founder variants: BRCA1_185delAG, BRCA1_5382insC and BRCA2_6174delT [6,7,8], whereas other BRCA1/BRCA2 pathogenic variants are rare [9,10]. Approximately half of these carriers do not have suggestive family history, and are not identified without a universal screening approach [3,11,12,13,14,15,16,17]. Lack of family history does not imply significantly lower cancer risks [3,11,18,19], and lifetime risk of developing cancer was 83%, 76% for BRCA1, BRCA2 carriers identified by screening, respectively, not significantly different than risks to carriers with family history. Once BRCA1/BRCA2 carriers are identified, there is an internationally recommended prevention and surveillance protocol [20]. Risk-reducing-salpingo-oophorectomy (RRSO), reduces OC by ~80%–95%, OC mortality by 80%, and overall mortality by ~70% in BRCA1/BRCA2 carriers [21,22,23,24]. The purpose of disease screening is to prevent disease by identifying persons at increased risk, or to improve prognosis by early detection.

The principles underlying disease screening have been formulated by Wilson [25], and are adapted for screening using genetic testing [26]. Although there are no universally agreed-upon criteria to qualify genes as appropriate for screening, genes should be clearly linked to disease, and justification regarding interventions with health benefits should be present. Implementing genomics-based screening programs wisely depends on ensuring that testing would be widely available and acceptable to the population, that disease penetrance be high in those screened positive, and that interventions would be cost-effective [26]. These principles are fulfilled by screening for BRCA founder variants among AJ. The advantage in BC prognosis has recently been demonstrated in a study performed by our group showing improved survival in women identified as BRCA carriers prior to BC diagnosis [27]. In AJ, the majority of BRCA1/BRCA2 carriers can be identified by an inexpensive test for founder variants, rather than full BRCA sequencing, which is performed for other populations.

In the general population, there are no effective methods for screening OC [28], and BC screening is based on early detection of cancer, i.e., secondary prevention, with mammography leading to improved prognosis [29]. BRCA1/BRCA2 screening offers an opportunity for primary prevention of cancer in unaffected carriers by risk-reducing mastectomy (RRM) and RRSO [21,22,24]. The remaining question for healthcare systems is whether BRCA1/BRCA2 screening in AJ is cost-effective. If cost-effective, it would fulfill all principles of disease screening [25], and implementation would be evidence-based [25].

The purpose of an economic evaluation is to assist policymakers in achieving optimal allocation of resources and maximizing social welfare. Cost-effectiveness analysis comparing relative health outcomes and costs of different strategies is the preferred form of economic evaluation, and quality-adjusted life year (QALY) is considered the most suitable determinant of health benefit, reflecting mortality and health-related quality-of-life effects. A systematic review [30] that assessed BRCA genetic testing programs in the USA and Europe, concluded that PS is cost-effective in AJ, given high carrier rate and low testing cost. As for the Israeli healthcare system, to date, only one economic analysis has been performed [31], but this was based on the entire Israeli population (rather than AJ). In almost all previous analyses, including the latter, cost-effectiveness was based on hypothetical models using parameters derived from the literature or empirical pricing. In this study, cost-effectiveness analysis was based on actual data from expenditures of a healthcare system. This real-life data achieves accuracy in comparing strategies.

Our objective was to assess the cost-effectiveness of PS for founder BRCA1/BRCA2 variants in all AJ women, as compared to testing based on two alternative comparators: A.International family history (IFH) criteria, a common strategy world-wide [17,20] requiring a probability of at least 10% for identifying a BRCA variant as indication for testing, based on acceptable prediction models [32,33].B.Cascade testing (CT), as outlined in Israeli Ministry of Health Guidelines (IMOH) directives [34], requiring probability of at least 25% for identifying a BRCA variant for testing. Women fulfilling these criteria are first/second degree relatives of known carriers. CT is an existing strategy implemented elsewhere, recently described as “an emerging strategy” [35].

## 2. Materials and Methods

The cost-effectiveness of PS was compared to CT and IFH. A decision tree was applied to each strategy, describing choices available to women offered genetic testing, accompanied by outcomes, probabilities and values, and probabilities were calculated based on previous studies. We set 30 years as lower bound for genetic testing since it is the age of surveillance initiation in carriers [20], and 65 years as an upper bound for genetic testing, since over age 65 surgical morbidity may increase, while effectiveness and cost-effectiveness may decrease.

Costs were based on MOH 2019 price list [36] and the study was conducted from the payer perspective; only direct costs were included. All costs were included, according to Clalit Health Services (Clalit) records (detailed below). Clalit do not own hospitals in the Jerusalem district, so estimates are likely to represent real-life costs. Costs were discounted at 3% per year. We used the medical component of Customer Price Index (CPI), published annually by the MOH, to adjust for inflation. In the past decade, the index has risen by about 3% per year. Prices were converted from New Israeli Shekels (NIS) to US dollars (USD) by exchange rate at time of analysis (mid 2019, 3.57 NIS per 1 USD).

QALYs were evaluated based on utility weights from the Tufts Cost-Effectiveness Registry [37], and life-years; QALYs were discounted. The incremental cost-effectiveness ratio (ICER) was calculated using the TREEAGE PRO 2022 software (TreeAge Software, Williamstown, MA, USA).

### Decision Tree Parameters

A.Identifying carriers:

Probabilities:

PS strategy: testing is offered to all AJ in the age range, so probability of testing is equal to uptake rate, which was estimated based on uptake rates in previous studies (67–71%) [12,16]. The probability of being identified as a carrier is the product of testing uptake and variant prevalence (2.5%), therefore is about 2.5% × 70% = 1.75%.

CT and IFH: testing is offered only to AJ fulfilling specific criteria. The probability of testing is the product of probability of being offered testing, and testing uptake. The probability of being identified as a carrier is the product of probability of testing and probability of carrier status for those tested.

Using the IFH strategy, the probability of being offered testing is 11% [11]. Uptake in this group was assumed to be complete, since these women are motivated to be tested, having been exposed to familial cancer and referred by physicians. The actual carrier rate in this group was estimated at 3.9%, based on our previous screening trial, making probability of being identified as a carrier 11% × 3.9% = 0.43%.

For CT, estimates are that only 10.9% of AJ carriers are being identified [38]. Thus, the frequency of identified AJ carriers in the AJ population is 1/367 (0.025 × 10.9% = 0.27%). In our previous study, each carrier had an average of five first or second-degree relatives, of whom half underwent testing [39]. Therefore, the rate of individuals whose risk is at least 25% and undergo testing is 0.68% (1/367 × 5/2). The actual rate of those tested in the study is 41.7% [40], since first-degree relatives are tested at double the rate of second-degree relatives (50% vs. 25% risk, respectively), making probability of being identified as a carrier 0.68% × 41.7% = 0.28%.

Costs:

In the PS arm, we estimated costs of a screening protocol, as in our previous study [11,12], in which women received written information pre-testing, and post-test in-person counseling was provided to carriers or women with significant family history. Cost of test was determined using actual large-scale testing cost, which is considerably lower per test. In the other two arms, costs were based on the MOH list for standard genetic testing process, which includes pre-test and post-test counseling.

B.Surveillance and prevention-probabilities and costs:

Probabilities of uptake of surgical risk-reducing measures were based on our BRCA carrier clinic [40], and data from the Israel Cancer Association Hereditary BC Consortium [41]. Surveillance adherence was assumed as complete. Surveillance and prevention costs were based on MOH prices. We used the carrier surveillance protocol recommended by the National Cancer Comprehensive Network (NCCN, USA) guidelines [20], for carriers aged 30–75, including: annual MRI, mammography, clinical breast exam, and biannual pelvic ultrasound, blood CA-125. After risk-reducing surgery, monitoring cost decreases. In Israel, carrier surveillance is included in provided health services. We assumed a mean age of carrier identification of 40 years, according to literature [42] therefore surveillance for approximately 35 years.

C.Cancer risks:

Probabilities of becoming affected with cancer are same in all models. For non-carriers, cancer incidence was taken from Israel Cancer Registry data [43,44] and literature [45]. For carriers, cancer risks data are based on studies about population-based carriers [11]. Since cancer risks in BRCA1 and BRCA2 carriers are different, probabilities of cancer diagnosis are weighted averages based on relative frequency of BRCA1 and BRCA2 carriers in AJ (~1:1.5).

D.Overall healthcare costs of women with BC, OC and unaffected:

Excluding limited reports [46], there are no substantial data on cancer cost in Israel. We therefore estimated this by using actual health expenditures in the Jerusalem district of Clalit. We received data for all women in Jerusalem that had been diagnosed with BC/OC between 1 January 2009 and 31 December 2010. The cost per year for affected women was derived from actual costs until 31 December 2016. The cost of healthy years for unaffected women was calculated based on the general capitation formula used for allocation of funds to health providers (HMOs) collected under National Health Insurance Law. The calculation was age-matched, i.e., we used capitation data for women ages 62–66 based on an average age of morbidity of 62 years and for five years following year of morbidity. The estimated overall healthcare costs of women with BC, OC and unaffected were 101,312 USD, 82,957 USD and 13,746 USD, respectively, similar to UK data [47].

E.Life expectancy and QALY:

Calculation of QALYs requires multiplication of utility weights for each health state and life expectancy of health states. Utilities were derived from literature and the Gertner Institute’s Technology Assessment Center and Tufts Medical Center Cost-Effectiveness Analysis (CEA) Registry database [37]. Life expectancy for unaffected women is based on lifetables from Israel Central Bureau of Statistics [48]. Life expectancy for affected women (BC or OC) was calculated as follows: healthy years from age 30 to mean morbidity age + sum of benefits for five years of illness + life expectancy at convalescence at full health. The QALY of affected women (BC or OC) was calculated using a model that assumes a mean age at morbidity and disease progression for five years, after which there is recovery or death; we then calculated a weighted average, accounting for the proportion of patients diagnosed at each disease stage. The mean ages used for BC and OC diagnosis were 43.8, 61.6 years and 55.6, 62 years in carriers and non-carriers, respectively. Survival data were collected from Israeli Cancer Registry and International NIH registry, according to stage of morbidity and number of years since diagnosis. The mean age at morbidity was calculated according to type of cancer and carrier status [43,49]. Estimated QALYs are shown in Appendix A.

F.Sensitivity analysis

We also performed one-way deterministic sensitivity analyses by calculating +25% for each parameter. The limits of most variables were abstracted from the medical literature [19]. Results of this analysis are presented using Tornado diagrams. We also performed probabilistic sensitivity analysis (PSA) using 1000 iterations. The results are presented using a cost-effectiveness acceptability curve and an incremental effectiveness scatterplot.

## 3. Results

### 3.1. Cost-Utility Analysis

Our analysis fulfills Consolidated Health Economic Evaluation Reporting Standards (CHEERS) (Appendix A). Detailed outcomes for costs, and QALYs for each probability in model, as well as comparisons between strategies regarding cancer incidence and cost-effectiveness, appear as Appendix A.

The decision tree that was applied to each of the three strategies for genetic testing, for comparing cost-effectiveness, describing choices available to women, outcomes, probabilities and values is presented in Figure 1. Probabilities were calculated based on previous studies (Table 1), and costs were based on MOH 2019 price list [36] (Table 2).

PS has the greatest effectiveness, and results in lifetime decreases of three BC and four OC cases per 1000 women vs. CT and one BC and one OC per 1000 women vs. IFH. The model predicted a gain of 21.6, 6.3 years per 1000 women, for PS vs. CT, IFH, respectively (Table 3).

Compared to IFH, the ICER per QALY for PS was 42,261 USD and compared to CT, the cost of IFH was −21,187.5 USD per QALY gained (Table 4). Although there is no official cost-effectiveness threshold in Israel to determine value for money, according to WHO criteria, strategies that cost less than once or triple the national annual GDP/capita are considered “highly cost–effective” and “cost–effective”, respectively, and annual GDP/capita in Israel is 36,250 USD [55]. Therefore, PS was not only the most effective, but also cost-effective.

### 3.2. Sensitivity Analysis

Figure 2 demonstrates comparison of PS to the other strategies. One-way sensitivity shows that the PS ICER is <7000 USD/QALY vs. CT even at extreme limits of all variables, which makes it highly cost-effective (Figure 2a). The PS is <100,000 USD/QALYvs. IFH (cost-effective) for extreme limits of most variables, with exception of carrier prevalence over ~3% and testing rates below ~65% (Figure 2b). PS is cost-effective even at current price of genetic testing, which is decreasing.

PSA using 1000 iterations to compare PS to the other strategies, presented as a cost-effectiveness acceptability curve (Figure 3) and Incremental effectiveness scatterplot (Appendix A), demonstrated that CT is dominated over all willingness to pay values, resulting from CT’s reduced effectiveness. PS has higher effectiveness, but also higher costs as compared with IFH. As the willingness threshold increases, more iterations result in PS being preferable.

The variables with most influence on ICER for PS are as follows, in descending order of magnitude:

PS vs. CT:

1. *Uptake and effectiveness of prevention.* Greater uptake of risk-reducing surgeries improves cost-effectiveness. BC risk was previously thought to be substantially reduced post-RRSO [21], however, later studies suggest minor reduction [54,56]. If RRSO-associated BC risk-reduction is < 25%, PS is not cost-saving, but remains cost-effective (Appendix A). Even if there is no RRSO-associated BC risk-reduction, the ICER for PS is 6788 USD/QALY (highly cost-effective).

2. *Surveillance cost in healthy carriers*. We compared ICERs of screening carriers from age 30, according to NCCN [20], vs. age 40, mean age of carrier identification assumed in model. If carriers are detected at age 30, the surveillance period is 10 years longer, and PS is not cost saving, though highly cost-effective (ICER/QALY = 2702 USD). Use of National Institute for Health and Care Excellence [NICE] protocol (the national cancer surveillance protocol in the UK for those at high risk of cancer) [57,58] would reduce PS ICER/QALY to −19,263 USD), increasing cost-effectiveness.

3. *Cancer risks and cost:* greater cancer risks and costs make PS more cost-effective. Recent extensive use of expensive PARP inhibitors increases this effect.

4. *Carrier prevalence*: Women choosing to participate in BRCA screening may not represent the AJ population, leading to higher or lower prevalence in PS. Carrier prevalence of 1.5–2.2% reduced the ICER/QALY of PS as compared with CT (Appendix A). At carrier rates >2.2%, although PS highly cost-effective, surveillance costs increase substantially.

PS vs. IFH:

1. *Carrier prevalence, rate of women offered testing*: Higher carrier prevalence led to higher costs in both strategies. However, when carrier rates in PS approach 3% or higher, PS cost-effectiveness is slightly lower compared to IFH. The larger the percentage of women being tested, the lower the PS ICER.

2. *Cancer risk*: higher cancer risks decreased the ICER for PS more than for IFH, since cancer is prevented in many more women in the former.

3. *Uptake and effectiveness of prevention*: greater uptake of RRM decreased the ICER for PS vs. IFH. Additionally, any RRSO-associated cancer risk-reduction increases PS vs. IFH cost-effectiveness.

4. *Surveillance cost in healthy carriers:* use of NICE would lower surveillance costs and decreased the ICER for PS vs. IFH, from 23,927 USD to 5089 USD.

## 4. Discussion

### 4.1. Main Findings

We carried out an economic evaluation for BRCA PS in unaffected AJ women, compared to two alternative strategies: “cascade testing” (CT), and “family history” (IFH). PS had the greatest effectiveness for cancer prevention, and includes a much broader population. PS results in a lifetime decrease of three BC cases and four OC cases per 1000 women vs. CT, a lifetime decrease of one BC and one OC per 1000 women vs. IFH, and a gain of 21.6 and 6.3 years per 1000 women for PS vs. CT and vs. IFH, respectively. PS was highly cost-effective vs. CT (ICER/QALY −3097 USD) and cost–effective compared to IFH (ICER/QALY 42,261 USD).

### 4.2. Comparison with Previous Studies

To the best of our knowledge there are no previous studies comparing cost-effectiveness of PS vs. CT, and none comparing three models. CT has recently been suggested as an alternative strategy [35], for identifying BRCA carriers at diminished costs while avoiding complexities of PS, such as accessibility to diverse populations, and counseling regarding variants of unknown significance (VOUS). However, the CT strategy is dependent both on the rate of identifying the first carrier in the family, and on familial communication. Current strategies only identify 10.9% of all AJ carriers [38], and less than half of relevant relatives undergo testing [39]. IFH misses half of women at risk for cancer [59,60,61] and prediction is not successful for all populations [62]. In the USA, only 10% of BRCA carriers were identified using IFH [52,63,64]. In an Israeli study, only two of five acceptable prediction-models predicted BRCA variants effectively [33], and none predicted BC effectively. Most carriers did not reach testing threshold, due to paternal inheritance, small families, lack of family history awareness, or inaccessible records. Cost-effectiveness of PS in other healthcare systems have been analyzed in a review, including studies from USA, UK, Norway, and Spain [30]. Three studies focused exclusively on testing for AJ founder variants [50,65,66]: Grann [65] and Rubinstein [66] analyzed PS for AJ in USA, the former demonstrated that PS would increase survival and be cost-effective if carriers performed risk-reducing surgery; the latter found that it was cost-effective regarding OC. However, both studies compared PS to no testing, rather than to another strategy, and the latter did not include BC in evaluation, a critical component. The third study [50] performed in UK, comparing PS to IFH, was similar to ours, and had similar results: PS was cost-saving, improved QALYs, and reduced incidence of OC (0.3%) and BC (0.6%). We are unaware of additional studies comparing PS to other models exclusively in AJ population. Two later cost-effectiveness analyses for PS vs. IFH in UK and USA were published by Manchanda, one on population of AJ and Sephardic Jews [47]; the other on Sephardic Jews [67]. Both showed that PS for founder variants was highly cost-effective. Meshkani [68] reviews twelve studies on BRCA testing in the general population. An additional study by Manchanda [59] compared PS to IFH, in multiple countries, and found that PS was not cost-effective in low-middle income countries, especially with low BRCA prevalence, but that PS was cost-effective in middle-high and high income countries It was recommended to further study cost-effectiveness.

### 4.3. Sensitivity Analysis

Uptake of risk-reducing surgery has major influence on PS cost-effectiveness, especially vs. CT. RRM rates in Israel are lower than many Western countries. [69] Since RRM has limited effect on mortality in BRCA carriers [70,71], increase in rate is unlikely. RRSO uptake in Israel is high, and change is unlikely to occur.

Cost of surveillance also had a large impact on cost-effectiveness. Our analysis was based on NCCN surveillance protocol. NICE recommendations limit MRI to ages 30–39 and mammography to 40–69 (vs. 30–75 for both by NCCN), reducing cost by 80%. Using NICE, PS would become even more cost-saving with ICER of −6857 USD, 19,263 USD, compared with CT, IFH, respectively. This is in accordance with previous analysis of PS in AJ16, based on NICE, that found ICER of −3521 USD/QALY for PS vs. IFH. Breast MRI accounts for 60% of surveillance cost, and continued decline of MRI cost would decrease surveillance cost, increasing PS cost-effectiveness. Although ovarian cancer surveillance has not been shown to be effective, we added its costs to our model, since until recently it was part of the surveillance protocol, and is still practiced in some centers. Exclusion of this cost would further increase cost-effectiveness. Our analysis was also conservative in that we did not assume lower costs of cancer care in cases where chemoprevention is effective, or where MRI surveillance results in downstaging.

The proportion of population offered BRCA testing, and carrier prevalence among those tested, had major influence on cost-effectiveness, especially vs. IFH. The proportion of women offered testing is most restrictive in CT (0.68%) [38], enlarged by IFH strategy (11%) [11] and highest for PS (67%) [12]. Testing a larger portion increases cost-effectiveness of PS vs. IFH. The lower the carrier prevalence among those tested, the more cost-effective PS becomes, compared to other strategies. If women who have partial AJ ethnicity are tested, we expect carrier rates to be lower than 2.5%. Albeit it is possible that in the first years of PS, carrier rates would be higher than 2.5%, similarly to our BRCA screening trial finding [12]. Nevertheless, although less cost-saving, PS remains highly cost-effective at higher carrier rates.

Lastly, PSA demonstrated that PS was the dominant strategy beyond WTP threshold of 42,500 USD, which is 1.17 times the GDP in Israel, making it cost-effective.

### 4.4. Strengths

In almost all analyses to date, cost-effectiveness of BRCA screening was based on hypothetical models using parameters from the literature or empirical pricing. Our evaluation is based on real-life expenditures. This direct data allows accuracy in comparison between strategies, and could be critical in prioritizing funding.

Another strength is our conservative estimates, which assumed higher cost of surveillance in unaffected women and lower costs of cancer care than real-life data in our population. Cost-effectiveness in reality is probably higher.

The model assumed 100% adherence of carriers to recommended cancer surveillance, and maximal cost for surveillance and risk-reducing surgery. As stated, the price of MRI is declining. Another conservative assumption was that risk-reducing surgery was performed immediately after carrier identification; therefore, costs were not discounted for later surgery performance, which may be more realistic. Besides surgery, there are emerging medical options to reduce BC in carriers, which may increase PS cost-effectiveness.

Cancer care is probably more expensive than our estimate, due to arrangements between HMOs and service providers. Price of OC treatment is rising due to extensive use of therapy such as PARP inhibitors. We did not include possibilities of relapse beyond five years. In reality, some women would relapse, raising cancer cost and reducing QALY. Therefore, we expect PS, which would prevent cancer, to have even better results per QALY. We calculated risk of either single BC or OC; the possibility of a second cancer in individual women was not considered.

All of these conservative estimates in our analysis underestimate cost-effectiveness of PS.

### 4.5. Limitations

Non-AJ populations have lower rates of BRCA1/2 variants. We did not estimate the cost-utility in “partially” AJ individuals, a growing population, due to inter-ethnic marriages. The cost of PS for women with partial AJ ancestry was higher than women of complete AJ background [20]. However, Manchanda [47] demonstrated that PS remained cost-effective if at least one grandparent was AJ, and cost-saving even if only two grandparents were AJ. Moreover, wider-range genomic PS in outbred populations has been found to be cost-effective [72]. Abu Husn [73] describes prevalence of BRCA1/2 pathogenic variants by Next Generation Sequencing (NGS) data from 30,223 individuals as 1:49, 1:81, and 1:103 in women of AJ, Southeast Asian non-AJ European ancestries, respectively, and reports that founder variants accounted for half of variants. Nevertheless, NGS is required for most populations.

PS remains cost-effective for a range of cost of testing for AJ founder variants (Appendix A), which is generally inexpensive. PS would be less cost-effective for the general population (BRCA or multigene panels) due to higher testing cost. Nonetheless, Manchanda [74] found that PS using cancer gene-panels in general UK & US population was more cost-effective than IFH. A later study showed that PS using multigene panels in general Australian population was highly cost-effective [75]. Therefore, PS will likely be cost-effective in our general population as well.

Another limitation is that our study is focused on the Israeli medical system, and is based on costs, surgical uptake rates and compliance with surveillance that are specific to Israel. These figures may not be accurate in healthcare systems elsewhere. Nevertheless, since implementation of PS has recently begun in Israel, this may form a basis for future comparison with medical systems involving additional populations.

We limited our analysis to women under 65 years of age, and it would be important to analyze cost-effectiveness for older women as well, as benefits will likely go beyond 65. Additionally, a societal perspective addressing productivity loss, was beyond the scope of our payer perspective analysis. However, this is again a conservative approach since accounting for productivity would increase cost-effectiveness.

### 4.6. Future Directions

PS in AJ has recently been initiated in Israel. This screening plan poses challenges: identifying population at risk, implementing surveillance program, and ensuring womens’ acceptability of surveillance. Future studies should focus on implementation of PS programs. Identification of population at risk requires finer tuning of ethnic origin. Our model included individuals of full AJ ancestry. Given social complexities of ethnicity-based testing, it is important to perform cost-analysis for women in general population. Cost-analysis of more general genetic PS (i.e., multigene panel testing) should be performed. In recent years, cost of NGS-based testing has declined. Although cost of NGS is still considerably higher than testing for founder variants, it is possible that for general PS, multigene tests would be more cost-effective than BRCA1/2 testing. NGS frequently reveals VOUS, which make result interpretation and counselling challenging. Previous studies have suggested not reporting VOUS [53]. The costs and management of VOUS need to be addressed in future studies on PS for the general population.

## 5. Conclusions

Our findings, based on actual expenditures from healthcare systems, indicate the advantage in expanding use of genetics for identification of women who are BRCA1/BRCA2 carriers, thus at high risk for developing cancer. The CT strategy, standard protocol till recently, severely restricts the number of carriers identified, and precludes effective surveillance and prevention in many carriers who remain unidentified. Compared to both CT and IFH policies, the PS strategy is cost-effective, and has the greatest gain in lifespan and reduction in cancer incidence. Implementation of PS for BRCA variants in AJ is the first example of using Precision Medicine for cancer screening. PS can also serve as an informative paradigm, as genomics is increasingly integrated into large-scale prevention for additional populations.

## Figures and Tables

**Figure 1 cancers-14-06113-f001:**
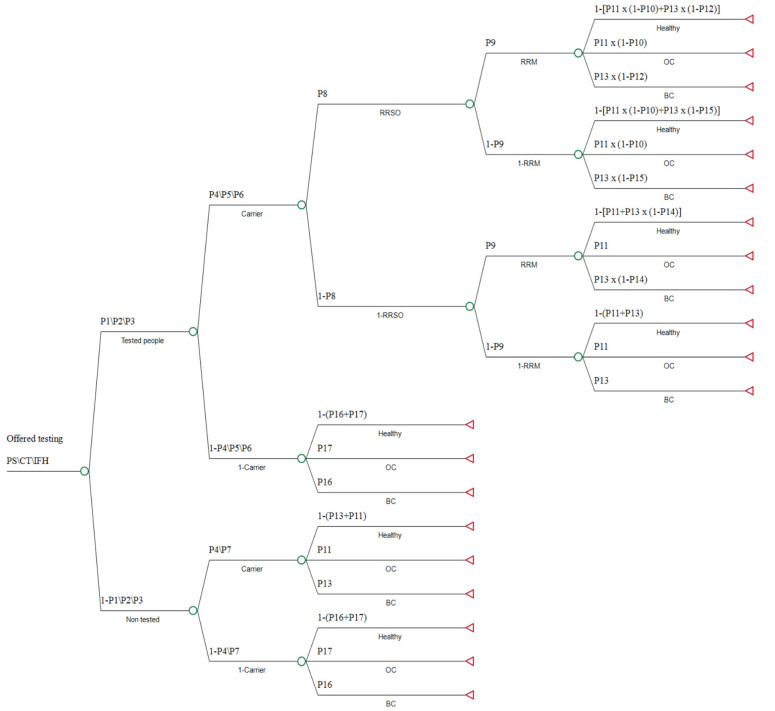
Decision tree. The decision tree has similar branches for all three strategies, and therefore shown only once. The numerical value of each probability (P) is shown in Table 1. In branches where different probabilities were used for each strategy, they are indicated in the following order PS\CT\IFH (PS—Population screening, CT—Cascade testing, IFH—International Family History). Healthy—unaffected with cancer. BC—Breast Cancer. OC—Ovarian Cancer. RRM—risk-reduction mastectomy. RRSO—risk-reduction salpingo-oophorectomy.

**Figure 2 cancers-14-06113-f002:**
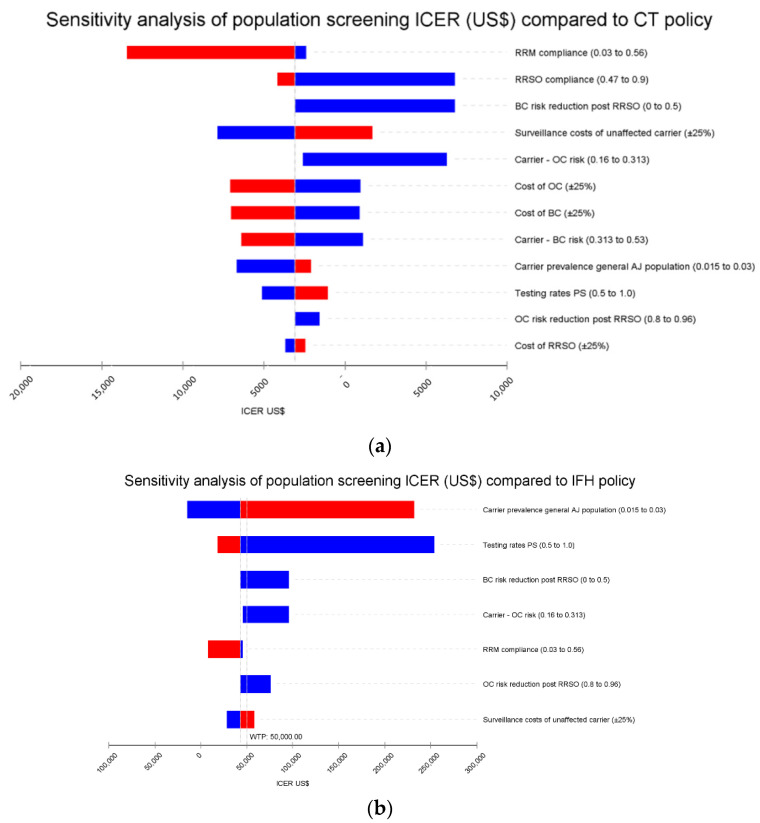
Sensitivity analysis comparing population screening to the other strategies. The results of the sensitivity analysis are shown as a tornado diagram. Variables and the range of values used in the sensitivity analysis are indicated to the right of the diagram. Red bars indicate the effect of increasing the variable (to the upper limit) on the ICER. Blue bars indicate the effect of decreasing the variable (to the lower limit) on the ICER. The vertical line (EV) indicates the ICER using the variable values in the original model. BC—Breast Cancer, OC—Ovarian Cancer, PS—Population screening, CT—Cascade testing, IFH–International Family-History based testing, RRM—risk-reduction mastectomy, RRSO—risk reduction salpingo-oophorectomy. (**a**) PS vs. CT, (**b**) PS vs. IFH.

**Figure 3 cancers-14-06113-f003:**
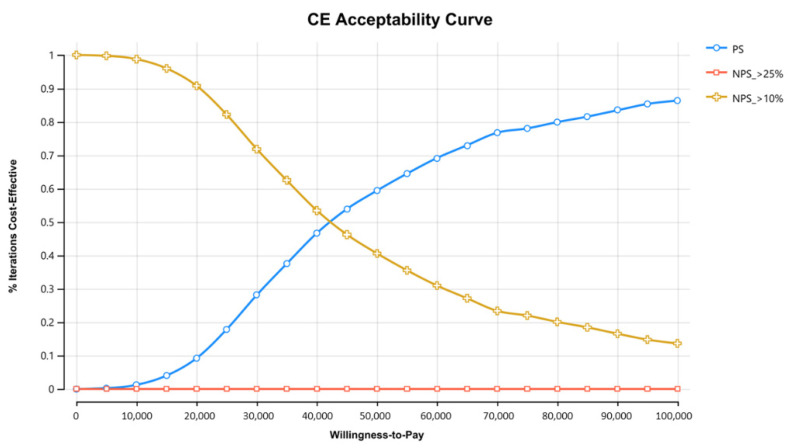
CE (Cost-effectiveness) acceptability curve comparing the three strategies. Probability Sensitivity Analysis using 1000 iterations to compare PS to the other strategies, demonstrated that CT is dominated over all willingness to pay values, resulting from CT’s reduced effectiveness. PS has higher effectiveness, but also higher costs as compared with IFH. As the willingness threshold increases, more iterations result in PS being preferable. PS- Population screening, CT—Cascade testing, IFH—International Family-History based testing.

**Table 1 cancers-14-06113-t001:** Probabilities in decision tree ^1^.

Variable		Value	Min	Max
**Testing rates**				
P1	Population screening (PS) [12]	67%	50%	100%
P2	Cascade testing (CT) [39]	0.68%	0.2%	1%
P3	IFH strategy [11]	11%	5%	15%
**Carrier prevalence**				
P4	General AJ population [11]	2.5%	1.5%	3%
P5	CT [39]	41.7%	25%	50%
P6	IFH-fulfilling testing criteria [12]	3.9%	2.45%	9%
P7	IFH-not fulfilling testing criteria [12]	1.7%	1.2%	2.04%
**Risk reducing surgery and cancer risk**				
P8	RRSO Compliance [41]	83.5%	47%	90%
P9	RRM Compliance	6%	3%	56%
P10	OC risk reduction post RRSO [21]	96%	80%	96%
P11	Carrier-OC risk [11]	31.3%	16%	30%
P12	BC risk reduction post RRSO & RRM [50,51]	95%	78%	99%
P13	Carrier-BC risk [11]	43%	31.3%	53%
P14	BC risk reduction post RRM [52,53] change 61 to ref from ref section listed as 63 (manchannd 2015), change 75 to ref from ref section listed as 22 (domchek 2010)	90%	44%	90%
P15	BC risk reduction post RRSO [52,54]	50%	0	50%
P16	Non carrier-BC risk [44,45]	13%	11%	14%
P17	Non carrier-OC risk [43,45]	1.5%	0.8%	1.5%

^1^ See methods for derivation of probabilities.

**Table 2 cancers-14-06113-t002:** IMOH price list for genetic counseling & testing, surveillance & risk-reducing surgery.

Health Service	Cost (NIS)	Cost (USD) ^1^
**Genetic counseling and testing**		
Genetic counseling	635	178
Genetic testing	635	178
Genetic testing-high throughput (PS ^2^)	80	22
**Carriers: Risk reducing surgery**		
RRM	29,650	8305
RRSO	16,110	4513
**Carriers: Surveillance modalities**		
CA-125 (twice a year)	96	27
Pelvic ultrasound (twice a year)	650	182
Mammogram (annual)	277	78
MRI (annual)	2060	577
Clinical breast examination (annual)	283	79
**Carriers: Total surveillance cost per year ^3^**		
Unaffected, no RRM/Affected with BC, post diagnosis	3366	943
Unaffected, post RRM	1029	288
Affected with OC, post diagnosis	2716	760
Affected with OC, post diagnosis & post RRM	379	106
**Carriers: Discounted cumulative surveillance cost (40–75 years) ^3^**		
Unaffected, no RRM/Affected with BC, post diagnosis	121,176	33,943
Unaffected, post RRM	37,044	10,376
Affected with OC, post diagnosis	107,526	30,119
Affected with OC & post RRM	23,394	6553

^1^ exchange rate at time of analysis (mid 2019, 3.57 NIS per 1 USD). ^2^ In the PS arm, cost of the genetic test is not based on the CT price list but was determined using actual costs of large-scale testing, which is considerably lower per test. All other costs in the table are from the CT price list. ^3^ We assumed that gynecological surveillance (pelvic ultrasound and CA-125) continues post-RRSO and that BC surveillance remines the same after BC diagnosis. Mean age of OC diagnosis were 55.6. BC—Breast Cancer, OC—Ovarian Cancer, RRM—risk reducing mastectomy, RRSO—risk reducing salpingo-oophorectomy.

**Table 3 cancers-14-06113-t003:** Lifetime rates of cancer in different testing strategies.

Lifetime Incidence Per 1000 Women by Testing Strategy			
Cancer	CT	IFH	PS
**Breast cancer**	138	136	135
**Ovarian cancer**	23	20	19

CT—Cascade testing, IFH—International Family-History based, PS—Population screening.

**Table 4 cancers-14-06113-t004:** Cost-effectiveness analysis of BRCA testing strategies.

Strategy	Cost (USD)	Incremental Cost (USD)	Effectiveness (QALY)	Incremental Effectiveness (Per 1000 Women)	ICER/QALY (USD)
**PS**	26,924		26.408		
**IFH (vs. PS)**	26,652	272	26.402	0.0063 (6.3 years)	42,261
**CT (vs. IFH)**	26,991	−339	26.386	0.016 (16 years)	−21,187.5

USD—United States dollars. IFH—International Family-History based. C—Cascade testing. PS—Population screening.

## Data Availability

Data is available per email contact with the corresponding author: amnonl@ekmd.huji.ac.il.

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
