# Peer review of "Real World Cost-Effectiveness Analysis of Population Screening for BRCA Variants among Ashkenazi Jews Compared with Family History-Based Strategies"

_cancers, 2022, doi:10.3390/cancers14246113_

Round 1
Reviewer 1 Report
This is a CEA of universal screening for BRCA variant among AJ women. I believe this is an important study and should provide many insights for BC screening in a specific population or precision medicine in future. However, I think there are some major comments on methodology to improve the quality and values of this manuscript.
Major
#Methodology
1. Improvement for the current health economic model
The author only used decision tree for CEA. The mainstay of CEA should be probabilistic sensitivity analysis (PSA), including cost-effectiveness acceptability curve (CEAC). PSA can incorporate more uncertainty although the author provided one-way analysis in this manuscript. The software of TreeAge can conduct PSA and CEAC.
2. I can understand the author want to compare PS vs. CT, and PS vs. IFH. Since there are three alternative polices here, I will suggest that each relevant strategy should be compared with the next best alternative in the cost-effectiveness plane, based on the economic concept of “opportunity costs".
3. The screening population is from 30-65 years of age. Do you mean: the genetic screening starts at age of 30 and annual mammography or MRI or CBE? I will suggest the author to explain more details for different scenarios.
4. QALY: based on supplementary table 1 and Page 4, I was unable to understand how the model calculate the QALY. Would the author elaborate it more.
#Results
When the policy with saving cost and better efficacy, it would be dominant. The author may not need to report the ICER ( with negative value)
#Minor
1. The captions of the tables in supplementary should be reviewed. It seems that there are so many repeats.
2. Suggest to mention the supplementary table number in the manuscript.
3. P8L276: what do you mean NICE protocol?
4. Since incidence rate of breast cancer is an outcome. The author may consider have CEA in terms of QALY and Life-expectancy.
In summary, this study is valuable for BC screening in AJ women. I would like to see the revised methodology for this important study.
Author Response
Responses to Reviewer 1’s comments:
Reviewer 1:
there are some major comments on methodology to improve the quality and values of this manuscript.
Major
#Methodology
- Improvement for the current health economic model: The author only used decision tree for CEA. The mainstay of CEA should be probabilistic sensitivity analysis (PSA), including cost-effectiveness acceptability curve (CEAC). PSA can incorporate more uncertainty although the author provided one-way analysis in this manuscript. The software of TreeAge can conduct PSA and CEAC.
Response to reviewer: Thank you for this helpful comment. In addition to the decision tree for CEA, we have now performed a PSA per your suggestion.
We have included the following explanation in methods section under the “sensitivity analysis” subtitle in lines 223-228: “We also performed one-way deterministic sensitivity analyses by calculating +25% for each parameter. The limits of most variables were abstracted from the medical literature [19]. Results of this analysis are presented using Tornado diagrams. We also performed probabilistic sensitivity analysis (PSA) using 1,000 iterations. The results are presented using a cost-effectiveness acceptability curve and an incremental effectiveness scatterplot.”
The results of PSA are presented using a CEA curve, which has been added as a new figure in the results section, lines 323-325 along with the following referral in the text, lines 316-321: “PSA using 1,000 iterations to compare PS to the other strategies, presented as a cost-effectiveness acceptability curve (Figure 3) and Incremental effectiveness scatter-plot (Supplemental Appendix 6), demonstrated that CT is dominated over all willing-ness to pay values, resulting from CT’s reduced effectiveness. PS has higher effective-ness, but also higher costs as compared with IFH. As the willingness threshold increas-es, more iterations result in PS being preferable.
The newly added PSA is also referred to in the revised discussion, section 4.3, lines 429-430: “Lastly, PSA demonstrated that PS was the dominant strategy beyond WTP threshold of 42,500 USD, which is 1.17 times the GDP in Israel, making it cost-effective.”
- I can understand the author want to compare PS vs. CT, and PS vs. IFH. Since there are three alternative polices here, I will suggest that each relevant strategy should be compared with the next best alternative in the cost-effectiveness plane based on the economic concept of “opportunity costs".
Response to reviewer: Thank you for this comment. To represent the comparison between alternatives, we changed table 4 accordingly (line 267). In the new version of table 4 the rows are ordered by effectiveness (top to bottom), from the most effective strategy (PS) to the least effective (CT). The data indicated in each row are now the incremental data compared to the nearest better alternative.
- The screening population is from 30-65 years of age. Do you mean: the genetic screening starts at age of 30 and annual mammography or MRI or CBE? I will suggest the author to explain more details for different scenarios.
Response to reviewer: Thank you for this comment. We have included the following revised explanation in lines 124-127 of the methods section of the manuscript: “We set 30 years as lower bound for genetic testing since it is the age of surveillance ini-tiation in carriers [20], and 65 years as an upper bound for genetic testing, since over age 65 genetic screening is less acceptableapplicable, as surgical morbidity increases, while effectiveness and cost-effectiveness decreases.”
The various scenarios are indicated in the decision tree (please see Figure 1 in manuscript, line 238).
As for surveillance, the protocol recommended starts at the same age, 30 years, and continues till 75 years, as described in methods section, lines 175-178: “We used the carrier surveillance protocol recommended by the National Cancer Com-prehensive Network (NCCN, USA) guidelines [42] (Supplemental Appendix), for carriers aged 30-75, including: annual MRI, mammography, clinical breast exam, and bi-annual pelvic ultrasound, blood CA-125.”
- QALY: based on supplementary table 1 and Page 4, I was unable to understand how the model calculate the QALY. Would the author elaborate it more.
Response to reviewer: Thanks for inquiring about how the model calculated the QALY. The revised explanation appears in line 202-221 of the methods section in the manuscript and is the following: “Calculation of QALYs requires multiplication of utility weights for each health state and life expectancy of health states. Utilities were, derived from literature and the Gertner Institute's Technology Assessment Center and Tufts Medical Center Cost-Effectiveness Analysis (CEA) Registry database [37]. Life expectancy for unaffected women is based on lifetables from Israel Central Bureau of Statistics [48]. Life expectancy for affected women (BC or OC) was calculated as follows: according to: healthy years from age 30 to mean morbidity age + sum of benefits for five years of illness + life expectancy at convalescence at full health. The QALY of affected women (BC or OC) The final QALY is the weighted average of Life expectancy multiplied by number of patients diagnosed at each disease stage. For women with BC, Quality of Life (Supplemental Appendix) was calculated using a model that assumes a mean age at morbidity and disease progression for five years, after which there is recovery or death; we then calculated a weighted average, accounting for the proportion of patients diagnosed at each disease stage. Life expectancy for unaffected women is based on lifetables from Israel Central Bureau of Statistics [48]. The mean ages used for BC and OC diagnosis were 43.8, 61.6 years and 55.6, 62 years in carriers and non-carriers, respectively. Survival data were collected from Israeli Cancer Registry and International NIH registry, according to stage of morbidity and number of years since diagnosis. The utility weight was multiplied by relative survival data. The mean age at morbidity was calculated according to type of cancer and carrier status [45,49].”
#Results
When the policy with saving cost and better efficacy, it would be dominant. The author may not need to report the ICER (with negative value)
Response to reviewer: We reported the ICER (with negative value) to allow an appreciation of the effect size, which we feel could be of practical valuable for policy makers.
#Minor
- The captions of the tables in supplementary should be reviewed. It seems that there are so many repeats.
Response to reviewer: Thank you for this helpful comment. We have reviewed and revised the captions of the supplementary tables to avoid this repetition.
- Suggest to mention the supplementary table number in the manuscript.
Response to reviewer: Thanks for this comment. I have mentioned the supplementary appendix numbers in the manuscript, in lines 221, 234, 333, 347, 467.
- P8L276: what do you mean NICE protocol?
Response to reviewer: Thank you for asking about this. The “National Institute for Health and Care Excellence [NICE] protocol” is the national cancer surveillance protocol in the UK for those at high risk of cancer” I have included this explanation in line 340-341 of the results section, and added two URLs to the protocols as new references (numbers 53,54). Our high-risk protocol was according to NCCN guidelines, and the NICE guidelines could be a possible alternative, which would actually improve CE, as discussed in lines 361-362 of the results section as well as lines 411-416 of the discussion section.
- Since incidence rate of breast cancer is an outcome. The author may consider have CEA in terms of QALY and Life-expectancy.
Response to reviewer: We considered performing CEA in terms of QALY and life-expectancy as you have suggested. However, the analysis presented here was originally intended for policy makers and were therefore honed, in attempt to provide focused results. We believe that the report, in its current structure, addresses the pertinent results for decision-making processes. We were concerned that including additional analyses with life-expectancy results would overcrowd the report.
In summary, this study is valuable for BC screening in AJ women. I would like to see the revised methodology for this important study.
Response to reviewer: Thank you for expressing your appreciation for the value of our study. We believe that the revised methodology improves the quality of the manuscript and trust that it addresses your concerns.
Reviewer 2 Report
Michaelson-Cohen et al have submitted a manuscript on a cost-effectiveness strategy to screen breast cancer mutants. Overall, the study is well executed and has the following suggestions for authors to consider in their revised version.
1. Encourage authors to rephrase the title of their manuscript as it has a huge similarity to Manchanda et al, 2014 PMID: 25435542 The current manuscript also utilizes family history relationships.
2. Please discuss the limitations of this approach.
Author Response
Response to Reviewer 2’s comments
Reviewer 2:
suggestions for authors to consider in their revised version.
- Encourage authors to rephrase the title of their manuscript as it has a huge similarity to Manchanda et al, 2014 PMID: 25435542 The current manuscript also utilizes family history relationships.
Response to reviewer: Thank you for pointing out the similarity of name to Manchanda’s paper. As you suggested, we have renamed the manuscript and used a name that stresses family history: “Real world cost-effectiveness analysis of population screening for BRCA variants among Ashkenazi Jews compared with family history-based strategies”
- Please discuss the limitations of this approach.
Response to reviewer: Thank you for inquiring about the limitations. In the previous limitations section, which discusses the higher cost expected in partial AJ and Non-AJ populations in lines 458-474: “Non-AJ populations have lower rates of BRCA1/2 variants. We did not estimate the cost-utility in “partially” AJ individuals, a growing population, due to inter-ethnic marriages. The cost of PS for women with partial AJ ancestry was higher than women of complete AJ background [20]. However, Manchanda [47] demonstrated that PS re-mained cost-effective if at least two grandparents were AJ. Moreover, wider-range genomic PS in outbred populations has been found to be cost-effective [69]. Abu Husn [70] describes prevalence of BRCA1/2 pathogenic variants by Next Generation Se-quencing (NGS) data from 30,223 individuals as 1:49, 1:81, and 1:103 in women of AJ, Southeast Asian non-AJ European ancestries, respectively, and reports that founder variants accounted for half of variants. Nevertheless, NGS is required for most popu-lations.
PS remains cost-effective for a range of cost of testing for AJ founder variants (Sup-plemental Appendix 5), which is generally inexpensive. PS would be less cost-effective for the general population (BRCA or multigene panels) due to higher testing cost. Nonetheless, Manchanda [71] found that PS using cancer gene-panels in general UK & US population was more cost-effective than IFH. Two later studies showed that PS us-ing multigene panels in general Australian population was highly cost-effective [53,72]. Therefore, PS will likely be cost-effective in our general population as well.”
In addition, we have added a more detailed description of the limitations of our approach. The revised limitations section appears in lines 475-479 of the discussion section in the manuscript and is the following:
“Another limitation is that our study is focused on the Israeli medical system, and is based on costs, surgical uptake rates and compliance with surveillance that are specific to Israel. These figures may not be accurate in healthcare systems elsewhere. Never-theless, since implementation of PS has recently begun in Israel, this may form a basis for future comparison with medical systems involving additional populations.”